# Self-Compassion during COVID-19 in Non-WEIRD Countries: A Narrative Review

**DOI:** 10.3390/healthcare11142016

**Published:** 2023-07-13

**Authors:** Yasuhiro Kotera, Ann Kirkman, Julie Beaumont, Magdalena A. Komorowska, Elizabeth Such, Yudai Kaneda, Annabel Rushforth

**Affiliations:** 1School of Health Sciences, University of Nottingham, Nottingham NG7 2HA, UK; 2College of Health, Psychology and Social Care, University of Derby, Derby DE22 1GB, UK; 3School of Medicine, Hokkaido University, Sapporo 060-8638, Hokkaido, Japan

**Keywords:** narrative review, non-WEIRD countries, self-compassion, COVID-19, mental health, WEIRD countries, cross-culture, self-compassion

## Abstract

The coronavirus disease 2019 (COVID-19) pandemic impacted people’s mental health negatively worldwide, including in non-WEIRD (Western, Educated, Industrialised, Rich and Democratic) countries. Self-compassion, kindness and understanding towards oneself in difficult times have received increasing attention in the field of mental health. Self-compassion is strongly associated with good mental health in various populations. This narrative review aimed to synthesise the evidence on self-compassion and mental health in non-WEIRD countries during the COVID-19 pandemic. MEDLINE and PsycINFO were searched for empirical studies. Self-compassion was consistently associated with positive mental health in non-WEIRD countries too. However, how, and to what degree, each component of self-compassion impacts mental health remains to be evaluated across different cultures. Future research such as multi-national intervention studies, or component network meta-analysis, is needed to advance our understanding of how self-compassion improves mental health in different populations.

## 1. Introduction

### 1.1. Mental Health Difficulties in COVID-19 Worldwide

Major epidemic outbreaks such as the coronavirus disease 2019 (COVID-19) pandemic pose an increased psychological demand on societies and individuals from many social groups around the world [1,2,3]. In 2022, the World Health Organization (WHO) reported a 25% increase in the prevalence of anxiety and depression following the pandemic, with women and young people being those most affected [4]. A systematic review investigating the impact of the pandemic on children and adolescents’ mental health across the world found that children in high school were most at risk for increases in worry and depression [5]. A cross-sectional study investigating the mental health outcomes of sexual and gender minorities (SGM; defined as non-cisgender and non-heterosexual people) in the USA found that SGM had significantly higher levels of depression and post-traumatic stress disorder (PTSD) symptoms due to COVID-related worries than non-SGM communities, possibly resulting from the increased mental health risks associated with these communities in pre-pandemic conditions [6]. Another group exposed to increased minority stress as a result of the pandemic is those from ethnic minorities. People identifying with Black, Asian and Minority Ethnic (BAME) groups experience significant mental health inequalities and difficulties in White-dominant societies [7]. Research assessing mental health changes in the United Kingdom (UK) using the UK Household Longitudinal Study data found that both women, regardless of ethnicity, and ethnic minority men experienced a greater increase in mental distress than White British men. Amongst the male population, Bangladeshi, Indian and Pakistani individuals experienced the highest increase in the UK [8]. Likewise, migrants reported substantial mental health difficulties during COVID-19, associated with various reasons including less developed social networks, language and cultural barriers, and financial and employment instability [9]. Additionally, people with disabilities, whose risk of death was three times higher than people without disabilities, experienced difficulties managing existing disability symptoms, accessing routine care and rehabilitation, and engaging with routine life activities due to restrictions during the pandemic [10]. In Italy, a highly infected country in Europe, a large-scale (*n* = 1827) study was conducted, spanning over four different time points during the pandemic (March 2020, August 2020, November 2020 and March 2021). The levels of perceived stress and state anxiety (a type of anxiety that is temporary, arising from a response to a certain event) were constant across those time points [11]. In the United States of America (USA), a significantly higher increase in symptoms of anxiety, depression, substance misuse and suicidal ideation was seen in people with disabilities than those without disabilities, with this population (i.e., people with disabilities) reporting that the pandemic had made it harder for them to access care and medication due to the stress on the healthcare system. That is, when healthcare providers are stressed, people with disabilities tend to struggle even more to access care and medication [12]. Despite the relatively low incidence of COVID-19 in Australia, there has been a significant lasting change in the mental health concerns of healthcare workers. Research measuring the longitudinal psychological impact of the pandemic on Australian hospital workers found that although scores on self-report mental health measures were significantly better than during the height of the pandemic, their mental health status had not returned to pre-pandemic levels, and that an increase in psychological distress was significantly more common among younger, early-career healthcare workers than more experienced ones [13]. These findings report that the COVID-19 impacted the mental health of diverse groups in many societies negatively.

### 1.2. Mental Health Difficulties in Non-WEIRD Countries

Historically, mental health research has focused on a small portion of the global population. People in Western, Educated, Industrialised, Rich, and Democratic (WEIRD) countries [14], who only account for 12% of the world population, represent 80% of social and behavioural science research participants [15,16]. The literature remains overwhelmingly WEIRD-focused rather than non-WEIRD-focused [17]. Non-WEIRD countries are those countries that do not apply to at least one of the five WEIRD characteristics. So, for example, countries like Japan are educated, industrialised, rich and democratic, however, are not Western. A systematic review on COVID-related depression and anxiety disorders around the world identified only seven studies from non-WEIRD countries out of forty-eight included studies [18]. Cultural differences in mental health and people’s behaviours have been reported for many years. Relating to COVID-19, cultural characteristics were relevant to people’s preventative behaviours [19]. Findings in one area of the world cannot be generalised into another area without adequate cultural adaptation [20]. Cultural adaptation means ‘the systematic modification of an evidence-based treatment (or intervention protocol) to consider language, cultural, and context in such a way that it is compatible with the client’s cultural patterns, meaning, and values’ [21]. This includes the consideration of spiritual beliefs, social norms, and the local context and local needs of the target population, which can help to improve service user attitudes towards treatment in the population [22]. The cultural adaptation of mental health treatment has been conducted in depth for some approaches, e.g., cognitive behavioural therapy [23] and mindfulness [24]. However, self-compassion-based treatment has not been evaluated nor discussed in depth, in terms of cultural adaptation. This suggests the importance of highlighting what has been found in non-WEIRD countries.

During the COVID-19 pandemic, the mental health difficulties of people in non-WEIRD countries were reported to some degree. For example, in Japan, high levels of stress, depression and anxiety were reported [25]. Increased rates of suicide and abuse were found, especially among young women [26,27]. Adolescent mental health was also a great concern and resilience was recommended as a coping strategy [28]. Migrants were also damaged mentally, in relation to economic/employment status [9]. A meta-analysis of the COVID-19 mental health impacts in Southeast Asian countries (Malaysia, Indonesia, Thailand, Vietnam, The Philippines and Singapore) identified a high prevalence of anxiety and depression (22% and 16%, respectively) [29]. In Singapore, social isolation was highlighted especially among children [30]. The prevalence of anxiety, burnout and depression increased from pre-COVID to during the pandemic among general practitioners [31]. A comparative study between older and younger adults identified that older ones had lower levels of depression, anxiety and stress and higher psycho-social adaptability than younger ones [32]. Interestingly, Singaporean mothers in low-income families were not affected markedly and the authors attributed it to resilience, which was supported by a social milieu they created using governmental financial support [33]. In China, 16.5% of people reported depression in the early stage of the outbreak [34]. This high level of depression was repeated after the peak of pandemic infection [35]. Social isolation was highlighted as a causal factor and contact with healthcare workers was identified as a key protective behaviour [36,37]. Moreover, COVID-19 survivors experienced depression and anxiety more than those who were not infected [38].

Turning to the Middle East and Africa, though the number of studies was modest, one study identified that 84.3% of people felt that their family relationships were unchanged or improved, whereas the remaining 15.7% reported that their family relationships became worse during the pandemic [39]. In the Arab Gulf region, though both groups scored highly, the prevalence of depression and anxiety symptoms was higher in people with diabetes than those without (61% of people with diabetes and 45% of people without) [40]. In the United Arab Emirates, foreign workers reported high rates of post-traumatic stress, depression, anxiety and insomnia, in particular among women, younger individuals and those with a previous diagnosis of a psychological disorder. Moreover, a positive correlation was found between pandemic severity in their home countries and their mental health symptoms [41].

In Africa, a meta-analysis identified a high prevalence of anxiety, depression and insomnia: the pooled prevalence was 37% in 27 studies, 45% in 24 studies and 28% in 9 studies, respectively. Mental health was poorer in North Africa than in Sub-Saharan Africa. This meta-analysis also highlighted that these prevalence rates were higher than in many other countries, calling for more COVID-19 mental health studies in Africa [42]. Another meta-analysis also reported a high prevalence of anxiety and depression in Africa, and identified that female gender and a history of medical conditions were major risk factors [43].

In South America, a Brazilian study reported high rates of anxiety and depression (52% and 40%) among adults who recovered from non-critical COVID-19 [44]. Among Brazilian men, high perceived stress and intolerance for uncertainty during the pandemic were strongly associated with mental health problems [45]. People in Chile experienced the COVID-19 pandemic when their mental status was still unsettled due to a social and political crisis that occurred towards the end of 2019 [46]. Markedly high rates of obsessive compulsive disorder were reported (48% during the pandemic, 2% before the pandemic) [47].

Although the number of studies was smaller than that from WEIRD countries, important findings were reported from non-WEIRD countries.

### 1.3. Self-Compassion

Self-compassion has attracted increasing attention in mental health research and practice [48,49,50]. Self-compassion is commonly regarded as kindness towards oneself in difficult times [51]. The understanding of self-compassion has evolved through research and with that of the founding construct of compassion [52]. Compassion, following the Buddhist philosophy, has been likened to the complete devotion and impulsive response of a mother in relieving her child of distress. Dalai Lama said ‘Compassion is the wish that others be free of suffering’ [53]. Gilbert illustrates compassion as an evolutionary motivation that sees an interactive flow, fostering compassion responsiveness, both within us and between ourselves and others [54]. The relationship between self-compassion and compassion creates much discussion, with the suggestion that further refined definitions, for each, may offer a more conclusive position on their connection [55]. A standardised five-component definition of compassion (see Table 1) is proposed to enable a more accurate comparison of the factors pertaining to both constructs [55].

Neff’s theoretical proposal for self-compassion [56], based on channelling the Buddhist philosophy of compassion towards the self, in conjunction with the Self-Compassion Scale [57], has enabled empirical research to thrive in this area. Self-compassion research started to be published in 2003 and it accounted for more than 4000 journal papers in 2021 [58].

This initial model for self-compassion consists of three main components: being kind to oneself, acknowledging a shared sense of humanity, and maintaining a state of mindfulness [56]. Progressing from this foundation, these three components are expanded to include bipolar elements: the absence of self-judgement, the requirement not to isolate oneself, and the avoidance of overidentification (excessively attaching oneself with their own thoughts and feelings and being swept away by them) (Table 1) [51]. Further distinctions are made for the three primary components of this self-compassion model between tender and fierce expression [59] (Table 2).

**Table 1 healthcare-11-02016-t001:** Comparison of models for compassion in five elements [55] and for self-compassion in six elements [58].

Compassion	Self-Compassion	
1. Recognising Suffering 5. Motivation to act/acting to alleviate suffering.	Self-Kindness (+)	(−) Self-Judgement
	Recognising one’s own behaviour, limitations and suffering, in the absence of harsh criticism, whilst actively caring for oneself and being motivated to alleviate distress.	
2. Understanding the universality of suffering in human experience 4. Tolerating uncomfortable feelings aroused in response to the suffering person (e.g., distress, anger, fear) so remaining open to and accepting of the person suffering	Common Humanity (+)	(−) Isolation
	Acknowledging one’s suffering to be integral to life and the shared experience of such suffering across humanity. A sense of togetherness through human imperfection.	
3. Feeling empathy for the person suffering and connecting with the distress (emotional resonance)	Mindfulness (+)	(−) Overidentification
	Maintaining a sense of awareness of the present situation and setting parameters of it, being in the present and not beyond its current scope.	

Achieving a balance between tender and fierce self-compassion expression is required to achieve a position that sustains both a healthy self-attitude, being able to have a positive self-regard and a shared sense of humanity, and an active response to distressing situations, creating boundaries and being compelled to take alleviating action [59]. When a balance is achieved between both tender and fierce expressions, self-compassion provides a safe and strong place to be present with difficulty or distress, whilst also maintaining the motivation to intervene and reduce suffering.

### 1.4. Study Aim

This narrative review aimed to identify evidence on self-compassion and mental health during the COVID-19 pandemic in non-WEIRD countries.

## 2. Methods

We decided to conduct a narrative review because this review allowed for a broader inclusion of studies than other forms of reviews such as systematic reviews [60]. As this review aimed to collect and synthesise the evidence from non-WEIRD countries that had not received great attention in research, the authors decided to employ a narrative review for the design.

A literature search was conducted to explore the evidence for self-compassion on mental health during the COVID-19 pandemic, focusing on non-WEIRD countries. Two databases, MEDLINE and PsycINFO, were used as these two are relevant to mental health and psychology research. The search strategy included terms such as ‘self-compassion’, ‘COVID-19’, ‘mental health’ and the locations of non-WERID countries. Publications in English between December 2019 and December 2022 were included.

As this was a narrative review, not a systematic review, relevant articles were reviewed by YK, AK, JB, MAK and AR. The key findings were thematically synthesised by YK to form a narrative about self-compassion on mental health during the COVID-19 pandemic in non-WEIRD countries, which was then reviewed by all authors, who agreed with the final version.

## 3. Results

### 3.1. Fewer Studies in Non-WEIRD Countries Than WEIRD Countries

Compared with self-compassion studies in WEIRD countries, the number of self-compassion studies in non-WEIRD countries during the pandemic was small. Self-compassion studies have been explored in many WEIRD countries during the COVID-19 pandemic, such as Spain [61], the USA [62] and the UK [63]. Some studies involved both WEIRD and non-WEIRD countries. For example, a cross-cultural comparison study between the USA, Japan, Malaysia and China found that people in all four countries showed an association between higher self-compassion and the prevention of negative mental health during the COVID-19 pandemic [64]. Moreover, studies using self-compassion interventions (not just as an outcome) were more limited in non-WEIRD countries. For instance, a systematic review of self-compassion therapies (from 2004 to 2020) found that most of the studies were conducted in WEIRD countries (USA = 8, UK = 4, Canada = 2, Netherlands = 2) and non-WEIRD countries were represented by only one study in Japan, China and Israel [65]. A systematic review exploring interventions in Arab countries found one self-compassion intervention study during COVID-19 [66]. This self-compassion intervention, offered online, reduced anxiety in 13 students through 10 sessions in four weeks [67]. As a pre-COVID systematic review on self-compassion reported, self-compassion research in non-WEIRD countries is scarce [68].

### 3.2. Consistent Positive Impacts of Self-Compassion on Mental Health in Non-WEIRD Samples

Non-WEIRD studies found that higher self-compassion supported positive mental health outcomes such as motivation, life satisfaction and wellbeing and decreased mental health problems. These studies suggested that people incorporate self-compassion practice in their daily life, including their workplace or school. For instance, self-compassion and amotivation (unwillingness to engage in work and a key mental health factor [69,70]) were negatively associated in Japanese workers (*n* = 165), indicating that workers with lower self-compassion are not autonomously motivated to work [71]. This relationship, self-compassion and amotivation, was also found in workers in South Africa [72]. Additionally, among Indonesian university students, autonomous motivation to learn during COVID-19 increased when the students were more self-compassionate [73]. Autonomous motivation predicted good mental health (reduces scores for depression, anxiety and stress) [74]. These studies suggested that incorporating self-compassion practice in daily life could help autonomous motivation, leading to better mental health.

Increased life satisfaction was positively associated with self-compassion among 337 self-quarantined individuals in China during the pandemic [75]. In Hong Kong (*n* = 761), individuals that were self-compassionate had less of an impact on their psychological distress from the perceived threats of COVID-19 [76]. Additionally, in the Indonesian population, self-compassion was a buffer for negative mental health and a preventive measure for the psychological negative impact of COVID-19 [77]. In Jakarta, the capital of Indonesia, self-compassion was identified as a key protective factor for positive wellbeing for unemployed graduates during the pandemic [78]. Likewise, higher self-compassion was positively associated with increased positive wellbeing and reduced stress during COVID-19 for Indian urban mothers (*n* = 242) [79]. Turning to Africa, first-year university students with higher self-compassion showed better adjustment to their university life in South Africa [80]. Self-compassion has been researched throughout non-WEIRD countries and has been found to have a positive association with better mental health during the COVID-19 pandemic. These studies concluded that interventions increasing self-compassion could be used to improve wellbeing and decrease psychological distress.

## 4. Discussion

This narrative review aimed to synthesise the evidence for self-compassion on mental health during the COVID-19 pandemic in non-WEIRD countries. The positive impact of self-compassion on mental health in non-WEIRD samples was consistent with findings from WEIRD samples. Our findings offer several research and practice implications.

### 4.1. Implications for Research and Practice

Self-compassion was helpful to people’s mental health in non-WEIRD countries during COVID. However, the quantity of studies was small in these countries. A pre-COVID cross-cultural study compared the relationships between self-compassion and cultural values across WEIRD and non-WEIRD countries [81]. This study identified significant relationships between self-compassion and some cultural values that are not often associated with WEIRD countries. For example, the positive aspects of self-compassion (i.e., self-kindness, common humanity and mindfulness) were associated with a cultural value of long-term orientation (i.e., focusing on the future rewards with an emphasis on perseverance and thrift [82]). Among the 11 countries included in this study, Korea, Japan and the UK were regarded as a long-term-oriented culture countries (>50 on the long-term orientation index). Korea and Japan are non-WEIRD countries. Moreover, self-compassion was negatively associated with a cultural value of indulgence: less indulgent, that is ‘more restrictive’ cultures tended to be self-compassionate. Egypt, Korea, Iran, Japan and Spain are restrictive culture countries (<50 on the indulgence index). All of these countries apart from Spain are non-WEIRD countries. The authors of this study discussed that these results can be partly explained by the fact that Eastern philosophy emphasises the practice of compassion. Indeed, self-compassion is highly influenced by Buddhism, which originated in non-WEIRD countries. However, self-compassion’s relevance to mental health has been observed across WEIRD and non-WEIRD countries [83,84,85,86,87]. These findings propose the mechanism of how self-compassion improves the understanding of mental health needs because how mental health and psychological trauma associated with COVID-19 are regarded can differ cross-culturally [30,88]. Moreover, cross-cultural differences in the self-compassion mechanism are worthy of investigation. For example, ‘self’ is an individualistic concept; therefore, self-kindness may be a key pathway of self-compassion, improving mental health in people in individualistic countries, and hence many WEIRD countries. On the other hand, among people in collectivistic countries, common humanity may be the key pathway. Further, the social aspect of compassion needs to be evaluated in different societies/communities. Multi-national intervention research or component network meta-analysis (i.e., a generalised version of network meta-analysis that estimates the effects of components of complex interventions [89]) is needed to evaluate these possible differences in self-compassion by culture.

In practice, self-compassion may help in understanding why certain interventions such as peer support have helped people’s mental health during COVID-19 [90]. Peers are those who have overcome mental health difficulties. In peer support work, peers meet with mental health service users, and offer emotional, social and practical support using therapeutic self-disclosure. Self-compassion has not been applied in peer support work. However, recently, self-compassion was proposed as a way to understand how peer support works [91]. Peer support workers talk about their own experiences regarding mental health difficulties to their service user who is currently undergoing similar mental health difficulties. This shared experience is a key component in peer support work [92]. From the self-compassion perspective, this can be understood as common humanity, meaning that the service user recognises that their peer support worker has also undergone similar challenges to theirs. This awareness helps to reduce their sense of isolation (e.g., “Why is this happening to me?” [93]), which in turn can lead to better mental health. In the context of COVID-19, mental health service users may feel a connection with their peer, which can help them to be more compassionate towards themselves, leading to better coping with distress during COVID-19. Practitioners can gain helpful insights into how their interventions help service user mental health by appraising it from a self-compassion perspective. More practice application needs to be conducted.

Another example of a mental health intervention showing that self-compassion can offer a new explanatory lens may be mental health recovery narratives [94]. Mental health service users engage with other people’s narratives about their mental health recovery. The service users feel hopeful (they too can overcome the difficulties) and more connected (awareness that they are not alone in experiencing the current difficulties). Similar to peer support, recovery narratives also offer service users stories of similar experiences to their mental health experiences. However, recovery narratives can take different forms such as texts, audio, video or art, and can be offered as a recorded format. If those narratives are stored online and accessible to many people, service users can access recovery narratives at any time they want from anywhere they want, as long as they have access to the internet. How recovery narratives help mental health may be understood from the self-compassion perspective. By engaging with others’ recovery narratives, a service user may feel that they are not alone; this can be more self-reassuring rather than self-criticising [95]. This can be framed as self-kindness in the self-compassion framework. The effects of mental health recovery narratives may be understood through the self-kindness pathway.

### 4.2. Limitations and Suggestions for Future Research

Limitations should be noted. Firstly, the literature we reviewed in this narrative review paper was limited to the English language. Considering that this narrative review is about non-WEIRD countries, other languages should be included. For example, searches in Malay might have offered relevant COVID-19 non-WEIRD studies, as this language is spoken in countries that are known for rapid academic output growth such as Malaysia [96]. However, English is used in many non-WEIRD countries and the research output of some English-speaking non-WEIRD countries is relatively high (e.g., India, Indonesia [97]). Therefore, the authors thought that the language by itself would not create a bias for study selection. Secondly, this was not a systematic review; therefore, some key papers might have been missed. However, we used established and relevant databases with search terms such as self-compassion and mental health. Additionally, we consulted a group of self-compassion researchers and practitioners for this review and no additional papers were identified by them. We conducted a narrative review for this topic because the advantages of a narrative review include having a broader scope, allowing for more flexible interpretation [98,99]. These advantages can offer helpful insights into this rather under-explored topic.

## 5. Conclusions

Self-compassion research in non-WERID countries during COVID-19 reported positive outcomes, as seen in research in WEIRD countries. Considering cross-cultural differences in mental health, the mechanism of how self-compassion has helped people’s mental health during and/or after the pandemic and how different it is cross-culturally need to be fully evaluated, theorised and methodologised. Moreover, appraising interventions used cross-culturally through a self-compassion lens may offer helpful insights. As self-compassion comprises three components, how these three components are enhanced in an intervention may offer one way to assess cross-cultural differences (i.e., from the same self-compassion intervention, as one component may be increased markedly in one culture, while another component may be increased markedly in another). The contentions discussed in this review can help to foster self-compassion research in non-WEIRD countries and can highlight areas for future self-compassion research.

## Figures and Tables

**Table 2 healthcare-11-02016-t002:** Suggested expressions of tender and fierce self-compassion for the three primary components of self-compassion by Neff [59].

Expressions of Self-Compassion			
Purpose	Self-Kindness	Common Humanity	Mindfulness
Tender (Be With)	Love	Connection	Presence
Fierce (Protect)	Bravery	Empowerment	Clarity
Fierce (Provide)	Fulfilment	Balance	Authenticity
Fierce (Motivate)	Encouragement	Wisdom	Vision

## Data Availability

Not applicable.

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
