# Peer review of "Self-Compassion during COVID-19 in Non-WEIRD Countries: A Narrative Review"

_healthcare, 2023, doi:10.3390/healthcare11142016_

Round 1
Reviewer 1 Report
The theoretical approach and methodology used is appropriate for a narrative review. It provides information relevant to this field of work. They are aware of the limitations of the type of review carried out. The format of some bibliographic references in the bibliography should be improved.
Author Response
Reviewer 1
Reviewer 1’s comment 1
The theoretical approach and methodology used is appropriate for a narrative review. It provides information relevant to this field of work. They are aware of the limitations of the type of review carried out. The format of some bibliographic references in the bibliography should be improved.
Authors’ response 1-1
Thank you for your feedback. Now the reference is re-formatted.
Reviewer 2 Report
Some countries are classified as Non-WEIRD countries just because of being Non-Western. However, they have conducted several studies on the topic of mental health during COVID-19. Therefore, it is difficult to conclude that Non-WEIRD countries have conducted less research on the discussed topic and generalizing findings.
Minor editing of English language required
Author Response
Reviewer 2
Reviewer 2’s comment 1
Some countries are classified as Non-WEIRD countries just because of being Non-Western. However, they have conducted several studies on the topic of mental health during COVID-19. Therefore, it is difficult to conclude that Non-WEIRD countries have conducted less research on the discussed topic and generalizing findings.
Authors’ response 2-1
If a country does not meet at least one of the five domains, that will be considered as a non-WEIRD country. E.g., Japan is educated, industrialised, rich and democratic, but not Western, thus a non-WEIRD country. So non-Western countries are non-WEIRD countries, and our searches followed that criterion. Apologies that we did not clarify this. This is now clarified in the manuscript (Section 1.2).
Reviewer 2’s comment 2
Minor editing of English language required
Authors’ response 2-2
Minor editing is done, thank you.
Reviewer 3 Report
The article supports the production of knowledge of interest to the scientific, professional, and political communities. It approaches the concepts of compassion and self-compassion and their impact on people's mental health, opening up a new analysis and reflection, outside the framework of self-esteem and empowerment, which seems very relevant to me.
It is well-written and well-grounded.
Even though it is a study with the limitations mentioned by the authors, it is relevant to our knowledge.
Author Response
Reviewer 3
Reviewer 3’s comment 1
The article supports the production of knowledge of interest to the scientific, professional, and political communities. It approaches the concepts of compassion and self-compassion and their impact on people's mental health, opening up a new analysis and reflection, outside the framework of self-esteem and empowerment, which seems very relevant to me.
It is well-written and well-grounded.
Even though it is a study with the limitations mentioned by the authors, it is relevant to our knowledge.
Authors’ response 3-1
Thank you for your feedback, capturing the strengths of our paper.
Reviewer 4 Report
I consider the article to be topical and very interesting, as studying the factors that may play a role in mental health in a pandemic helps us to prevent future pandemics. You never know. The introduction to the article is very well written and clear. It is also very well justified and documented with relevant literature. The objective is well specified for a narrative review and I share the delimitations of the study. However, I think that important elements are missing in the method, in particular:
- The number of articles selected for the review is not specified. Nor is there a flow chart of the study selection process. I think these elements are important to be able to replicate the study.
- The table used to extract the data from the different articles is not indicated, nor are the elements to be recorded.
- Without these elements, in addition to being difficult to replicate the study, it is also difficult to know whether the results obtained are valid or not.
- I believe that the bibliography should indicate the articles used for the study.
Author Response
Reviewer 4
Reviewer 4’s comment 1
The theoretical approach and methodology used is appropriate for a narrative review. It provides information relevant to this field of work. They are aware of the limitations of the type of review carried out. The format of some bibliographic references in the bibliography should be improved.
Authors’ response 4-1
Thank you for your feedback. As the journal accepts papers in a free format at the first instance, we did not follow the MDPI style. Now the reference is re-formatted.
Round 2
Reviewer 2 Report
The aim of this narrative review was to synthesise the evidence on self-compassion and mental health in non-WEIRD countries during the COVID-19 pandemic, However, the Authors stated “Although the number of relevant studies was smaller than those from WEIRD countries” / “There was less evidence reported in non-WEIRD countries than in WEIRD countries”; these statements cannot be concluded from a narrative review. To make such conclusion, a systematic review must be conducted.
Conclusion should be revised according to the aim of the study.
Author Response
Thank you for your helpful feedback. In line with your comment, now our statements regarding the quantity of articles are removed.